# Transcriptomic Profiling for Prognostic Biomarkers in Early-Stage Squamous Cell Lung Cancer (SqCLC)

**DOI:** 10.3390/cancers16040720

**Published:** 2024-02-08

**Authors:** Maja Šutić, Branko Dmitrović, Antonia Jakovčević, Feđa Džubur, Nada Oršolić, Željko Debeljak, Asta Försti, Sven Seiwerth, Luka Brčić, Goran Madzarac, Miroslav Samaržija, Marko Jakopović, Jelena Knežević

**Affiliations:** 1Laboratory for Advanced Genomics, Division of Molecular Medicine, Ruđer Bošković Institute, 10000 Zagreb, Croatia; maja.sutic@irb.hr; 2Department of Pathology, Faculty of Dental Medicine and Health Osijek, Clinical Medical Center Osijek, 31000 Osijek, Croatia; branko.dmitrovic@fdmz.hr; 3Department of Pathology, School of Medicine, University of Zagreb, 10000 Zagreb, Croatia; antonia.jakovcevic@kbc-zagreb.hr (A.J.); sven.seiwerth@mef.hr (S.S.); 4Clinical Department for Respiratory Diseases Jordanovac, University Hospital Centre Zagreb, School of Medicine, University of Zagreb, 10000 Zagreb, Croatia; fedja.dzubur@kbc-zagreb.hr (F.D.); miroslav.samarzija@kbc-zagreb.hr (M.S.); 5Division of Animal Physiology, Faculty of Science, University of Zagreb, 10000 Zagreb, Croatia; nada.orsolic@biol.pmf.hr; 6Clinical Institute of Laboratory Diagnostics, University Hospital Center Osijek, 31000 Osijek, Croatia; zeljko.debeljak@kbco.hr; 7Faculty of Medicine, J.J. Strossmayer University of Osijek, 31000 Osijek, Croatia; 8Hopp Children’s Cancer Center (KiTZ), 69120 Heidelberg, Germany; a.foersti@kitz-heidelberg.de; 9Division of Pediatric Neurooncology, German Cancer Research Center (DKFZ), German Cancer Consortium (DKTK), 69120 Heidelberg, Germany; 10Diagnostic and Research Institute of Pathology, Medical University of Graz, 8010 Graz, Austria; luka.brcic@medunigraz.at; 11Department for Thoracic Surgery, University Hospital Zagreb, 10000 Zagreb, Croatia; goran.madzarac@kbc-zagreb.hr; 12Faculty of Dental Medicine and Health, J.J. Strossmayer University of Osijek, 31000 Osijek, Croatia

**Keywords:** NSCLC, squamous cell lung cancer (SqCLC), profiling, mRNA, biomarkers, survival, tumor microenvironment (TME), T cells

## Abstract

**Simple Summary:**

Lung cancer is a leading cause of cancer-related deaths worldwide, characterized as a disease usually diagnosed in the advanced stages with limited therapeutic options. Significant progress in the treatment of lung cancer has been achieved by the application of targeted therapy and immunotherapy based on the identification of molecular abnormalities of the tumor tissue. However, only a small proportion of patients is carrying genetic alterations and will benefit from targeted therapy. Here, we conducted transcriptomic profiling and a comprehensive biostatistics analysis of the squamous-cell lung carcinoma (SqCLC), a histological subtype of the non-small cell lung carcinoma (NSCLC), aiming to identify the specific transcriptomic signature of SqCLC and evaluate the functional relevance of identified genes. This study sheds light on individual SqCLC tumors’ transcriptomic landscape and discusses the therapeutic and prognostic potential of identified biomarkers.

**Abstract:**

Squamous cell lung carcinoma (SqCLC) is associated with high mortality and limited treatment options. Identification of therapeutic targets and prognostic biomarkers is still lacking. This research aims to analyze the transcriptomic profile of SqCLC samples and identify the key genes associated with tumorigenesis, overall survival (OS), and a profile of the tumor-infiltrating immune cells. Differential gene expression analysis, pathway enrichment analysis, and Gene Ontology analysis on RNA-seq data obtained from FFPE tumor samples (*N* = 23) and healthy tissues (*N* = 3) were performed (experimental cohort). Validation of the results was conducted on publicly available gene expression data using TCGA LUSC (*N* = 225) and GTEx healthy donors’ cohorts (*N* = 288). We identified 1133 upregulated and 644 downregulated genes, common for both cohorts. The most prominent upregulated genes were involved in cell cycle and proliferation regulation pathways (MAGEA9B, MAGED4, KRT, MMT11/13), while downregulated genes predominately belonged to immune-related pathways (DEFA1B, DEFA1, DEFA3). Results of the survival analysis, conducted on the validation cohort and commonly deregulated genes, indicated that overexpression of HOXC4 (*p* < 0.001), LLGL1 (*p* = 0.0015), and SLC4A3 (*p* = 0.0034) is associated with worse OS in early-stage SqCLC patients. In contrast, overexpression of GSTZ1 (*p* = 0.0029) and LILRA5 (*p* = 0.0086) was protective, i.e., associated with better OS. By applying a single-sample gene-set enrichment analysis (ssGSEA), we identified four distinct immune subtypes. Immune cell distribution suggests that the memory T cells (central and effector) and follicular helper T cells could serve as important stratification parameters.

## 1. Introduction

Lung cancer is a leading cause of cancer-related mortality, accounting for approximately 20% of all cancer-related deaths worldwide [1]. Despite recent advances in treatment options, in terms of immunotherapy and targeted therapy, 5-year survival remains poor. The highest survival rates of 30% are recorded in developed countries like Japan [2], while developing and/or low-income countries struggle with survival rates of less than 10% [3]. Lung cancer is traditionally divided into small-cell lung carcinoma (SCLC) and non-small-cell lung carcinoma (NSCLC). SCLC is less frequent than NSCLC, which accounts for 85% of all lung cancer cases. Histologically, NSCLC is further classified into adenocarcinoma, squamous cell carcinoma (SqCLC), large-cell carcinoma, and other not-so-common subtypes [4]. Successful treatment of lung cancer is complex and heavily dependent on the stage at the time of diagnosis, histologic subtype, patient clinical status, and comorbidities [5].

The usage of high-throughput technologies on genomic, transcriptomic, proteomic, or epigenetic levels has improved our understanding and characterization of the molecular pathogenesis of cancers, which has led to the identification of specific cancer cell vulnerabilities and triggered the development of targeted therapeutics. For example, genomic profiling of lung adenocarcinoma has led to the identification of many targetable alterations like mutations in the EGFR gene or rearrangements of the ALK gene, which can now be treated with specific targeted drugs [6,7,8]. High-throughput sequencing studies on NSCLC have, therefore, not only revealed the genomic landscape of this disease [9,10,11,12] but also identified genomic differences between the two most common NSCLC subtypes, adenocarcinoma and SqCLC [13]. Most importantly, targetable mutations identified in adenocarcinoma are rarely found in SqCLC patients. Despite growing efforts in further characterization of the genomic landscape of SqCLC, translation of acquired genomic knowledge into the development of targeted therapeutics for SqCLC has been much slower than in adenocarcinoma, resulting in only a few approved therapeutics. The discovery of immune checkpoint inhibitors was the next revolution in the therapy of NSCLC, with the programmed cell death protein-1/programmed cell death ligand-1 (PD-1/PD-L1) pathway as the most utilized one. Inhibition of this pathway enables priming and activation of the anti-tumor activity of cytotoxic T cells [14]. It has been shown that SqCLC patients can benefit from treatment with immunotherapy if their lung tumors express PD-L1 in 50% of tumor cells [15], while those who are not eligible are most likely to be treated with platinum-based chemotherapy as a first-line therapy.

In addition to genomic characterization, profiling of the tumor tissue at the transcriptomic level also has led to new insights into the tumorigenesis of this disease. So far, profiling of SqCLC transcriptome has allowed the identification of several SqCLC subtypes, based on gene expression level, that differ in biological processes and affect patients’ survival [16,17], indicating its usefulness in prognosis estimation.

Tumor cells are in constant interaction with the elements of the tumor microenvironment (TME). TME consists of an extracellular matrix and stromal, vascular, endothelial, and immune cells. Within TME, cancer cells can reprogram stromal cells infiltrating tumors to promote tumorigenesis [18]. It has been reported that the level of immune cell infiltration into the tumor depends on the tumor stage [19,20]. Chronic inflammation in lung cancer can also affect immune cell differentiation [21], which could lead to imbalanced antitumor activity, tumor evasion [17], or resistance to the therapy with immune checkpoint inhibitors [22,23]. The level of tumor-infiltrating immune cells also has prognostic value. For example, increased infiltration levels of B cells, T cells, and dendritic cells have been associated with a better prognosis for NSCLC. In comparison, increased infiltration levels of regulatory T cells and tumor-associated macrophages have been associated with a worse prognosis for NSCLC [24]. A high neutrophil-to-lymphocyte ratio has also been associated with poor overall survival and progression-free survival in NSCLC [25]. Since it has been shown that the profile of tumor-infiltrated immune cells could influence the clinical outcome of NSCLC patients [26], further characterization of the tumor microenvironment in SqCLC is also needed.

Since SqCLC is a complex disease, it is of crucial importance to enhance our understanding of its biology and genetic profile. The results of those studies could be used to enhance SqCLC management. Despite the growing number of mRNA sequencing studies, the number of comprehensive analyses of mRNA expression profiles of SqCLC is limited. Therefore, this study aimed to contribute to the current knowledge of the molecular background of SqCLC tumorigenesis by analyzing expression profiles of mRNA in SqCLC to explore the molecular background of SqCLC tumorigenesis.

## 2. Materials and Methods

### 2.1. Study Design

The whole transcriptome sequencing of formalin-fixed, paraffin-embedded (FFPE) SqCLC samples, coupled with FFPE healthy control tissue, was performed. Additionally, various transcriptomic profile analyses, including the identification of differentially expressed genes, Gene Ontology, and gene-set enrichment analysis, were performed. The level of immune cells infiltrating tumor tissue was also estimated. The sequencing results of the TCGA (The Cancer Gene Atlas)-LUSC and GTEx (Genotype-Tissue Expression) cohorts were included to validate differentially expressed genes identified in the experimental cohort. Finally, the impact of expression levels of validated differentially expressed genes on patients’ overall survival was examined. The systematic workflow of this study is graphically presented in Figure 1.

### 2.2. Patient Samples and Data Collection

Samples of primary SqCLC were collected at the University Hospital Centre Zagreb, Department of Thoracic Surgery and Department for Respiratory Diseases, Jordanovac, during surgical resection. A total of 23 tumor samples included in this study were taken between 2013 and 2019. Demographic and clinical characteristics of the tested population are summarized in Table 1. In addition, the three healthy control samples were collected at the Department of Anatomy, Histology, and Pathology, Faculty of Dental Medicine and Health, University of Osijek. Control tissue was obtained during autopsies of healthy individuals who had died from accidental deaths and did not have any lung changes, as confirmed by clinical pathologists. Both healthy and tumor specimens were fixed in formalin and embedded in paraffin. From each FFPE tumor sample, two 4 µm thick slices and four 10 µm slices were cut. The 4 μm sections were stained with hematoxylin and eosin (H&E) and reviewed by the pathologist at the Department of Pathology and Cytology, University Hospital Centre Zagreb. Study inclusion criteria for FFPE lung tumor specimens were histologic diagnosis of SqCLC, at least 60% of tumor cells, and not more than 30% of necrotic tissue in the sample. In addition, clinical data, like smoking status, TNM status, stage, and survival data, were collected. All raw data generated in this study, including FASTQ sequencing data and metadata, have been submitted to the NCBI Gene Expression Omnibus (GEO; https://www.ncbi.nlm.nih.gov/geo/ (submitted on 19 March 2023) under accession number GSE230089 https://www.ncbi.nlm.nih.gov/geo/query/acc.cgi?acc=GSE190089 (submitted on 19 March 2023).

For validation of our transcriptomic data, we downloaded an expression dataset named TcgaTargetGtex_gene_expected_count (version 2016-09-03) from the UCSC Toil RNA-seq recomputed compendium using the UCSC Xena browser. This dataset is composed of uniformly realigned expression data from TCGA (The Cancer Genome Atlas), TARGET (Therapeutically Applicable Research to Generate Effective Treatments), and GTEx (Genotype-Tissue Expression) samples [27]. Gene expression RNA-seq data were downloaded as log2(expected_count+1), together with phenotype data (TcgaTargetGTEX_phenotype.txt). We included 513 samples in the validation cohort, 225 squamous cell lung cancer patients, and 288 healthy controls. To ensure that the validation cohort is similar to the tested cohort, we set several inclusion criteria. Patients should be Caucasians; patients should be diagnosed with squamous cell lung cancer; patients should not have prior treatment; the site of resection or biopsy should be the lung (main bronchus and overlapping lesion of the lung were removed). Patients who have synchronous malignancy or had prior malignancy were not included in this study. We also excluded TARGET data since they are based exclusively on pediatric data, as well as TCGA healthy tumor adjacent lung samples. A list of all TCGA samples included in this study, together with clinical and demographic data, can be found in Appendix A and is summarized in Table 1.

### 2.3. RNA Isolation, Library Preparation, and RNA-seq Analysis

For the transcriptome profiling analysis of SqCLC and healthy control tissues, we extracted total RNA, prepared libraries, and performed RNA-seq. Total RNA was extracted from 2× 10 µm FFPE tissue slices with an RNEasy FFPE kit (Qiagen, Hilden, Germany), according to the manufacturer’s instructions. RNA integrity and quality were assessed using RNA Nano 6000 BioAnalyzer chips (Agilent, Santa Clara, CA, USA), with DV200 measurement setup (the percentage of fragments >200 nucleotides). Samples with DV200 values >30% were included in this study. The TruSeq RNA Exome kit (Illumina, Cambridge, UK) was used for library preparation, according to the manufacturer’s protocol, excluding the fragmentation step. Sizes and concentration of final libraries were estimated with High Sensitivity DNA chips on BioAnalyzer (Agilent, Santa Clara, CA, USA). Prepared libraries were sequenced on the Illumina HiSeq4000 platform to obtain 100-bp paired-end reads. The quality of sequenced reads was checked with the FASTQC application on the Base Space Sequence Hub cloud (Illumina, Cambridge, UK). Reads were then aligned to the human genome (NCBI GRCh38 Decoy) with the RNA-Seq Alignment App (Illumina, Cambridge, MA, USA).

### 2.4. Differential Gene Expression Analysis

In differential expression analysis, 23 SqCLC and 3 healthy samples were included, with a count depth of >40 million reads. Differentially expressed genes (DEGs) of SqCLC compared to healthy controls were determined using RNA-Seq Differential Expression Application (Illumina) that uses the DESeq2 tool. Differentially expressed genes were determined based on|log2FC|≥1 and adjusted *p* < 0.05, according to Benjamini and Hochberg’s method. To validate the results obtained on the experimental cohort, consisting of a relatively small number of tissue samples, expression profiling was also conducted on a validation cohort. We performed differential gene expression analysis using TCGA LUSC cohort (*n* = 225) and GTEx healthy controls (*n* = 288) from UCSC Toil RNA-seq data collection. Differential expression analysis on the validation cohort was performed in R using the DESeq2 package. For comparison of gene expression levels between two cohorts, we set |log2FC|≥ 1. A Venn diagram was plotted for visualization in the R using the Venn diagram package.

### 2.5. Gene Ontology (GO) Analysis and Gene Set Enrichment Analysis (GSEA)

The biological significance of differentially expressed genes was explored using Gene Ontology analysis. This analysis was performed using the web-based tool available at http://geneontology.org/ and PANTHER complete annotation sets for biological process (BP), cellular component (CC), and molecular function (MF) categories (GO Ontology database released 1 July 2022). We performed two separate GO analyses, one for upregulated and the other for downregulated genes. For statistical analysis purposes, we chose the Fisher’s Exact test, coupled with false discovery rate correction. GO terms with FDR q-values < 0.05 were considered significant. Results are presented as bar charts created in the R program (v. 3.6.2.) with the ggplot2 package. Gene set enrichment analysis of gene expression data was performed with desktop tool GSEA (v4.0.3) and MSigDB C2, all canonical pathways gene set collection. Analysis was run as a weighted pre-ranked list, based on log2 FC values, with 1000 permutations, max size = 500, and min size = 10. Results are presented as bar charts created in the R program with the ggplot2 package.

### 2.6. Survival Analysis

We performed a survival analysis to investigate the correlation of the patient’s overall survival (OS) with expression levels of commonly dysregulated genes in both cohorts. Since the experimental group is relatively small, for survival analysis, we used nonmetastatic patients’ data from the validation cohort. Death was considered an event of interest, and data for patients who did not die during follow-up were censored. Analysis was performed using the Xena browser. For each gene of interest (commonly dysregulated in both cohorts), patients were stratified into two groups based on the gene expression level (high or low expression). Cut-off values for allocating patients to either group were derived from the median expression level of corresponding genes. The Kaplan–Meier and log-rank tests were used to evaluate the difference in survival rates between the two groups. A *p*-value of less than 0.05 was considered significant.

### 2.7. Estimation of Tumor-Infiltrating Immune Cells

These analyses aimed to better define the tumor microenvironment in the context of the tumor-infiltrating immune cells. For an initial approximation of the level of tumor-infiltrated immune cells, we used the ESTIMATE algorithm (Estimation of Stromal and Immune cells in Malignant Tumors using Expression data) in the R estimate package. ESTIMATE is based on the single-sample gene-set enrichment analysis (ssGSEA) method that uses gene signatures related to stromal tissue and immune cells and outputs stromal score (SS), immune score (IS), and combined SS and IS as ESTIMATE score for each sample in analysis. To gain better insight into immune cell types that infiltrate the tumor, we performed an additional analysis, GSVA (Gene Set Variant Analysis), as described here [28], using 16 immune cell populations representing adaptive and innate immune systems. We included gene set signatures from Bindea et al. [29] for B cells, mast cells, macrophages, immature and activated dendritic cells, Neutrophils, NK dim, NK bright, T effector cells, T central memory cells, T helper, T follicular helper cells, and Cytotoxic cells. Gene sets representing cytotoxic cells included combined genes over-expressed in activated CD8+ T cells, T gamma delta cells, and NK cells. For CD8+ T cells, T gamma delta cells, and T regulatory cells, we used genes from Charoentong et al. [30]. A list of the genes used for GSVA enrichment can be found in Appendix A. Using GSVA scores of 16 immune cell populations and cytotoxic cells, coupled with hierarchical agglomerative clustering (Euclidian distance and Ward’s linkage), immune subtypes were identified. For both algorithms, rlog-transformed expression profiles were used as input. The immune cell infiltration analysis was not performed on the TCGA data because we did not have access to the FASTQ files of the selected samples, and, therefore, we found it inappropriate to directly compare identified subtypes in different sets of data.

## 3. Results

### 3.1. Study Populations

In our study, we used biological FFPE samples (gathered in the experimental cohort) to analyze the transcriptome profile of SqCLC. To validate our results, we included a validation cohort made of patients with similar clinical data to those in the experimental cohort. Demographic and clinical data of all patients involved in our study show that patients in both groups are of similar age (64 and 67 years, respectively) and are predominantly men. Smoking habits are also similar in both cohorts since the majority of patients in both groups are ex-smokers, with a similar percentage of current smokers in both cohorts. It is important to say that the majority of patients in both cohorts were diagnosed as nonmetastatic (experimental cohort (91%) and validation cohort (80%)). All available demographic and clinical data of the SqCLC patients in the experimental and validation cohorts are presented in Table 1.

### 3.2. Functional Characterization of Differentially Expressed SqCLC-Specific Genes

To gain insight into the gene expression profile contributing to the SqCLC phenotype in the experimental cohort, we conducted a comprehensive differential gene expression analysis using the DESeq2 tool. In this analysis, we identified a total of 2887 differentially expressed genes, comprising 1620 upregulated genes and 1267 downregulated genes in SqCLC tumors when compared to healthy controls (Figure 2A). The most significantly upregulated genes were MAGEA9B (log2 FC = 22.82, q-value = 1.7^−7^), STAG3L3 (log2 FC = 22.07, q-value = 1.13^−12^), CSAG1 (log2 FC= 21.68, q-value = 2.35^−8^), MAGED4 (log2 FC= 11.52, q-value = 6.26^−9^), CST1 (log2 FC = 10.83, q-value = 4.28^−5^), CACNA1B (log2 FC = 10.75, q-value = 1.09^−7^), MAGEA6 (log2 FC = 9.75, q-value = 2.26^−3^), PRAME (log2 FC = 9.58, q-value = 2.13^−8^), POU6F2 (log2 FC = 9.53, q-value = 2.66^−6^), and ZIC2 (log2 FC = 9.37, q-value = 4.54^−7^). The most significantly downregulated genes were DEFA1B (log2 FC = −14.21, q-value = 2.94^−7^), CEACAM8 (log2 FC = −9.9, q-value = 7.17^−3^), DEFA1 (log2 FC = −9.89, q-value = 1.06^−3^), S100A12 (log2 FC = 9.21, q-value = 3.3^−6^), MMP8 (log2 FC = −9.08, q-value = 9.54^−7^), MS4A3 (log2 FC = −9.03, q-value = 3.16^−8^), CELA3A (log2 FC = −8.99, q-value = 4.71^−3^), DEFA3 (log2 FC = −8.98, q-value = 8.03^−3^), MYL2 (log2 FC = −8.35, q-value = 1.11^−2^), and CYP1A1 (log2 FC = −8.32, q-value = 3.65^−4^). A detailed list of all differentially expressed genes in the experimental cohort (up and downregulated) can be found in Appendix A.

Next, we conducted a Gene Ontology analysis (GO) to gain more insight into the biological functions associated with the differentially expressed genes. The GO functional annotations provided valuable insights into which biological processes (BP), molecular functions (MF), and cellular components (CC) were enriched in SqCLC. Our analysis revealed that among the upregulated genes, there was an over-representation of genes primarily associated with the regulation of the cell cycle and proliferation (BP) (cell cycle (GO:0051276, q-value = 6.34^−25^), cell cycle process (GO:0007049, q-value = 6.24^−24^), regulation of cell cycle process (GO:0048285, q-value = 2.4^−16^), regulation of cell cycle (GO:0006259, q-value = 1.09^−14^), regulation of transcription factors (MF)(DNA-binding transcription activator activity (GO:0001216, q-value = 2.99^−3^), DNA-binding transcription activator activity, RNA polymerase ii-specific (GO:0001228, q-value = 3.66^−3^), and components related to the chromosomal/nuclear region of the cell (CC) (chromosome GO:0005694, q-value = 3.42^−18^), chromosomal region (GO:0098687, q-value = 1.58^−13^), chromosome, centromeric region (GO:0000775, q-value = 5.11^−12^), and condensed chromosome (GO:0000793, 4.55^−9^) (Figure 2B). Among the downregulated genes, the over-representation was mainly linked to immune system functions (Response to stimulus (GO:0050896, q-value = 3.31–21); Response to external stimulus (GO:0009605, q-value = 9.09–18); defense response (GO:0006952, q-value = 8.46–16); immune system process (GO:0002376, q-value = 5.58–15), including leukocyte activity (myeloid leukocyte activation (GO:0002274, q-value = 2.79^−9^), and cellular secretion processes (BP), regulation of receptor activity (MF), and components of the extracellular space, plasma membrane, and vesicles (CC) (Figure 2C). Detailed results for all Gene Ontology categories can be found in Appendix A.

To gain deeper insights into the underlying biological processes associated with dysregulated genes and to retrieve the phenotypic differences and the pathways they are involved in, we performed a Gene Set Enrichment Analysis (GSEA). The results of the GSEA analysis revealed an over-representation of pathways related to the keratinization (q-value < 0.001), NP63 pathway (q-value = 4.2^−4^), activation of ATR in response to replication stress (q-value = 5.25^−4^), FOXM1 pathway (q-value = 7.05^−4^), cell cycle checkpoints (q-value = 9.13^−4^), PLK-mediated pathways (q-value = 0.004), and AUROA B pathway (q-values = 0.004). Conversely, under-representation was observed in immune-related pathways such as the TLR signaling pathway (q-value < 0.001), neutrophil degranulation pathway (q-value < 0.001), triggering of the complement regulation of the NFĸB activation (q-value = 0.008), immunoregulatory interactions between lymphoid and nonlymphoid cells (q-value = 0.017), and the leukocyte transendothelial migration pathway (q-value = 0.016). The overview of the most significant enriched pathways is shown in Figure 2D, and a detailed list of all identified pathways is listed in Appendix A.

### 3.3. Validation of DEGs on TCGA SqCLC Cohort

Since the experimental cohort utilized in this study was relatively small in size, we sought to validate our findings on a larger sample set. Therefore, we conducted a validation analysis using the TCGA-LUSC cohort (N = 225) and healthy controls from the GTEx dataset (N = 228). To compare gene expression levels between these two cohorts, we established a criterion of |log2FC|≥ 1. First, we analyzed differentially expressed genes (DEGs) in the validation cohort, and we identified that 5678 genes were upregulated, while 3992 genes were downregulated (Figure 3A). The most prominent upregulation was observed for KRT16 (log2 FC = 11.65, q-value < 1.0^−300^), LINCO1206 (log2 FC = 11.53, q-value < 1.0^−300^), KRT6B (log2 FC = 11.33, q-value < 1.0^−300^), CALML3 (log2 FC = 11.23, q-value < 1.0^−300^), DSG3 (log2 FC = 11.12, q-value < 1.0^−300^), PADI3 (log2 FC = 11.03, q-value < 1.0^−300^), MAGEA9B (log2 FC = 10.63, q-value < 1.0^−300^), UGT1A7 (log2 FC = 10.60, q-value < 1.0^−300^), S100A7 (log2 FC = 10.41, q-value < 1.0^−300^), and TMPRSS11D (log2 FC = 10.28, q-value < 1.0^−300^) genes. On the other side, the most prominent downregulation was observed for DEFA1B (log2 FC = −8.95, q-value < 1.0^−300^), DEFA3 (log2 FC = −8.74, q-value < 1.0^−300^), DEFA1 (log2 FC = −8.57, q-value < 1.0^−300^), DEFA4 (log2 FC = −8.08, q-value = 5.41^−298^), PRTN3 (log2 FC = −7.79, q-value < 1.0^−300^), IL1RL1 (log2 FC = −7.15, q-value < 1.0^−300^), ELANE (log2 FC = −6.99, q-value < 1.0^−300^), ADAMTS7P3 (log2 FC = −6.75, q-value < 1.0^−300^), MYH7 (log2 FC = −6.68, q-value < 1.0^−300^), and AMY2A (log2 FC = −6.59, q-value = 4.05^−147^) genes. A detailed list of all DEGs, both up- and downregulated in the TCGA-LUSC cohort, are provided in Appendix A. Next, we analyzed commonly dysregulated genes specific to both tested cohorts and found that 1133 genes were consistently upregulated in both cohorts, while 644 genes were downregulated (Figure 3B). The most prominent upregulation in both cohorts, experimental and validation, was observed for MAGEA9B (log2 FC = 22.82 vs. log2 FC = 10.63), MAGED4 (log2 FC = 11.52 vs. log2 FC = 3.88), CST1 (log2 FC = 10.83 vs. log2 FC = 9.93), CACNA1B (log2 FC= 10.75 vs. log2 FC = 6.61), PRAME (log2 FC = 9.58 vs. log2FC = 9.32), POU6F2 (log2 FC = 9.53 vs. log2 FC = 7.01), ZIC2 (log2 FC = 9.37 vs. log2 FC = 7.04), B4GALANT4 (log2 FC = 9.30 vs. log2 FC = 4.41), and CALML3 (log2 FC = 9.21 vs. log2 FC = 11.23) genes. The most prominent downregulation in the experimental and validation cohort was observed for DEFA1B (log2 FC = −14.21 vs. log2 FC = −8.95), CEACAM8 (log2 FC = −9.9 vs. log2 FC = −6.10), DEFA1 (log2 FC = −9.89 vs. log2 FC = −8.57), S100A12 (log2 FC = −9.21 vs. log2 FC = −2.56), MMP8 (log2 FC = −9.08 vs. log2 FC = −3.19), MS4A3 (log2 FC = −9.03 vs. log2 FC = −5.09), DEFA3 (log2 FC = −8.98 vs. log2 FC = −8.74), MYL2 (log2 FC = −8.35 vs. log2 FC = −5.63), CYP1A1 (log2 FC = −8.32 vs. log2 FC = −4.33), ALB (log2 FC = −8.25 vs. log2 FC = −3.62), APOA2 (log2 FC = −8.16 vs. log2 FC = −1.55), and DEFA4 (log2 FC = −8.04 vs. log2 FC = −8.08) genes. The common DEGs found in both cohorts are provided in Appendix A.

Gene Ontology (GO) analysis showed that the upregulated genes in the TCGA cohort were predominantly associated with the regulation of cell cycle signaling tissue development (anatomical structure development (GO:0140014), q-value = 3.17^−12^, tissue development (GO:0051983), q-value = 9.24^−11^, DNA replication (GO:0051301, q-value = 1.36^−12^), chromosome segregation (Nuclear chromosome segregation (GO:1903047, q-value = 1.92^−18^), chromosome segregation (GO:0098813, q-value = 6.07^−18^), sister chromatid segregation (GO:0010564, q-value = 1.01^−15^), cell division (Mitotic cell cycle (GO:0022402), q-value = 1.86^−22^, mitotic cell cycle process (GO:0000278, q-value7.79^−19^), nuclear division (GO:0007059, q-value = 6.22^−18^) (BP), protein binding, transmembrane signaling receptor activity, molecular transducer activity (MF), and membrane-enclosed lumen, protein-containing complex, membrane-bounded organelle and cytoplasm (CC). GO analysis for downregulated genes in the validation cohort revealed a predominant association with the regulation of the cellular component movement (GO:0051270, q-value = 8.93^−18^), cell migration (regulation of cell migration (GO:0030334, q-value = 1.14^−17^), regulation of cell motility (GO:2000145, q-value = 1.41^−17^), regulation of locomotion (GO:0040012, q-value = 6.62^−17^), regulation of response to stimulus (GO:0048583, q-value = 7.18^−16^), surface receptor signaling pathway (GO:0007166, q-value = 2.88^−14^), cell adhesion (GO:0007155, q-value = 1.78^−13^), cell communication (BP) (GO:0007154, q-value = 4.03^−13^), nucleic acid binding, signaling receptor binding, immune receptor activity, DNA and RNA binding, cytokine binding (MF), and components of the plasma membrane, cell periphery, nucleolus, extracellular matrix, secretory granule, immunoglobulin complex, and intracellular protein-containing complex (CC) (Figure 3C,D). Detailed results for all Gene Ontology categories are listed in Appendix A. When we analyzed functional annotations of commonly dysregulated genes in both cohorts, we observed a similar pattern of consistently up- and downregulated genes and impacted pathways. Also, we did not perform GSEA for the validation cohort.

### 3.4. Survival Analysis

Given the high mortality rate of the patients diagnosed with SqCLC and lung cancer patients in general, it is extremely important to identify prognostic biomarkers for those patients. In our pursuit to predict if the DEGs could serve as prognostic biomarkers for overall survival (OS) in the SqCLC patients, we conducted a survival analysis on genes that were consistently upregulated in both cohorts. Aiming to identify prognostic biomarkers associated with OS for early-stage SqCLC patients, only patients diagnosed with nonmetastatic disease were included. We assessed the impact of genes with a log2FC ≥ |1| on patients’ prognosis, specifically focusing on OS. Utilizing the online platform Xena and the TCGA validation cohort, we employed the Kaplan–Meier survival analysis and the log-rank test. Based on median expression levels, we categorized samples into either high- or low-expression groups. A significance level of *p*-value less than 0.05 was considered statistically significant. Our analysis revealed that 36 differentially expressed genes exhibited a significant correlation with patients’ OS. Among these, 20 genes were associated with worse OS, while 16 genes were associated with better OS, i.e., could be considered protective. Identified genes, indicated as associated with worse/better OS, are listed in Appendix A. Among the genes associated with worse OS, several exhibited a highly significant impact on patients’ OS (*p*-value < 0.01). Notably, HOXC4 (*p*-value = 0.0001), LLGL1 (*p*-value = 0.0015), SLC4A3 (*p*-value = 0.0034), RNFT2 (*p*-value = 0.0047), and CEP72 (*p*-value = 0.0063) demonstrated the most pronounced effects (Figure 4A). Overexpression of these genes was associated with poor overall survival. Overexpressed genes associated with better/prolonged OS (*p*-value≤ 0.01) were GSTZ1 (*p*-value = 0.0029), ENTPD3 (*p*-value = 0.001), DEPDC1 (*p*-value = 0.0019), and GRHL3 (*p*-value = 0.0023) (Figure 4B).

### 3.5. Estimation of Immune Cell Infiltration Level in SqCLC

Our results on differentially expressed genes indicated that the downregulation of the most prominent genes is associated with the regulation of the immune response. As already mentioned in the introduction part of the manuscript, immune cells are an important component of the TME. Therefore, we decided to elucidate TME in the context of the immune cell profile. For initial TME exploratory analysis, we choose ESTIMATE since it gives a general overview of immune cells infiltrating tumor tissue. Our results showed that there is a difference in tumor-infiltrating immune-cell levels among tested tumor samples (Figure 5A). Since initial TME analysis showed a difference in the level of immune cells infiltrating the tumor tissue, we performed an additional analysis, GSVA, to investigate specific types of immune cells that are infiltrating the tumor tissue. Based on calculated GSVA enrichment scores, we grouped our SqCLC samples into four immune subtypes (Figure 5B). Out of 23 SqCLC samples, 8 were clustered in subtype 1 (34%), 7 were clustered in subtype 2 (30%), 3 were clustered in subtype 3 (13%), and 5 were clustered in subtype 4 (21%). Considering the main characteristics of the immune cells that dominantly infiltrate individual subtypes and the level of infiltration, it was shown that subtypes 1 and 2 could be described as less immunogenic, in contrast to subtypes 3 and 4, designated as more immunogenic. The stratification criteria were established based on the identified cell-specific markers (see Materials and Methods section). Less immunogenic samples exhibit decreased levels of infiltration with NK^bright^ and CD8+ T cells, while more immunogenic samples exhibit higher levels of infiltration with central memory T cells, effector memory T cells, and follicular helper T cells compared to subtypes 1 and 2. Detailed immune cell profiles are presented in Figure 5C.

## 4. Discussion

Since the molecular-targeted approach in the diagnosis and treatment of SqCLC is very limited in clinical settings, an increasing number of studies are focused on the identification of specific diagnostic and therapeutic biomarkers. Due to the genomic heterogeneity of individual SqCLC tumors, it is extremely important to characterize the biological features of as many individual samples of SqCLC as possible, both at the genomic and transcriptomic levels and thus contribute to a better understanding of the mechanistic background of the disease and support development of the new therapeutic solutions. Our study aimed to analyze expression profiles of mRNA in SqCLC tumor samples, with the ultimate goal of identifying the key genes and pathways associated with tumorigenesis and prognosis.

Analysis of differentially expressed genes, commonly deregulated in both experimental and validation cohorts, and their functional annotation (characterization) demonstrate several interesting results. The most upregulated genes belong to the MAGE gene family (MAGEA9B, MAGED4), emphasizing their relevance in SqCLC. The melanoma antigen gene (MAGE) protein family is a group of proteins normally expressed only in reproductive tissues, while their abnormal expression is observed in various types of tumor tissues. Due to their unique immunogenic nature and expression restricted to tumor tissues, they have been suggested as diagnostic biomarkers [31]. Overexpression of MAGE genes has been associated with poor outcomes, like decreased survival in NSCLC patients [31]. In contrast to other members of the MAGE family, MAGED4 demonstrates low expression levels in many tumor tissues, while its expression in NSCLC is relatively high and significantly higher in SqCLC than in adenocarcinoma [32]. All these results demonstrate the importance of further research on the MAGE family, both as a prognostic and therapeutic biomarker.

In our cohorts, significant enrichment of the genes involved in the keratinization and the formation of keratinized layers in tumor tissue (KRT5, KRT6A-C, KRT13-KRT17, KRT19) and genes associated with the regulation of cell division (FOXM1, p63) was identified. It is known that keratin filaments play a role in forming the protein structural framework within the epithelial cells and protecting them from different types of stressors [33]. Keratins are also a characteristic feature of early-stage SqCLC, while reduced in lung adenocarcinoma [34], and are extensively used as specific diagnostic markers for different tumor types, including squamous cell carcinomas, which are characterized by keratinocyte hyperproliferation [35]. Another interesting finding is FOXM1 overexpression, a transcription factor that plays a critical role in normal lung development, affecting the differentiation and function of lung epithelial cells. FOXM1 is also important for cell cycle regulation as it controls the transition from the G1 to S phase, G2 to M phase of the cell cycle, and progression through the M phase of the cell cycle. Several critical regulators of mitosis, such as Skp2, cyclin A and B, Cdc25A, Cdc25C, AURKB, BIRC5, and CENPA, are under the transcriptional control of the FOXM1 gene [36], and all these genes were overexpressed in our tested cohorts, as also shown by others [37,38]. Furthermore, our GO analysis identified that upregulated genes that participated in the DNA repair are highly enriched. They play a crucial role in genome integrity control and protection, and their deregulation is a hallmark of tumor aggressiveness. There are several mechanisms associated with DNA repair, and our results indicated excessive activation of the several genes responsible for maintaining genome integrity. For example, E2F transcription factors are involved in cell cycle control by regulating genes responsible for the transition from G1 to the S phase, showing increased expression in our cohorts. Based on the results of several studies, E2F subtypes, including E2F8, can be considered oncogenes contributing to the development of the squamous subtypes of lung tumors [39,40,41]. Our results confirm the findings of these studies, showing that E2F8 is overexpressed in both cohorts and emphasizing its potential role in SqCLC progression.

Significantly downregulated genes detected with DEG analysis were associated with the regulation of immune response. We identified defensins, like DEFA1B, DEFA1, DEFA3, and DEFA4, among the most significantly downregulated genes that are involved in the regulation of immune response [42]. In general, defensins are small proteins with strong antimicrobial and immunomodulatory functions. They are expressed predominantly in neutrophils and epithelial cells, exhibiting strong cytotoxic activity directed to cancer cells. All these characteristics make them potential chemotherapeutic targets [43,44]. The DEFA1B gene belongs to the α-defensin subgroup, which is known to promote tumor cell proliferation, contributing to tumor progression and invasiveness [45], as well as influencing tumor microenvironment due to their chemotactic properties (CD4+, CD8+, immature DC, monocytes) [46]. They also exhibit strong effector activity by triggering degranulation of the polymorphonuclear cells, including neutrophils, suggesting that they have a role in tumor-directed immune activity. Interestingly, we identified significantly reduced neutrophil degranulation. Neutrophils are among the first immune cells recruited to the site of inflammation, and their abundance in tumor tissue usually correlates with a worse prognosis [47]. Evidence shows that neutrophils can act as partners of tumor cells in cancer progression, making them interesting anticancer targets. It is worth mentioning that the plethora of genes that potentially could participate in the degranulation process was downregulated in our study. For example, CXCR1 and CXCR2 are receptors expressed on neutrophils. They can be activated with different cytokines and chemokines expressed by tumor or endothelial cells, resulting in either recruitment of the neutrophils to the tumor tissue or pro-angiogenic processes [48,49]. It has been shown that pharmacological inhibition of CXCR1 or CXCR2 leads to the promotion of the antitumor T cell response as a consequence of the limited neutrophil tumor infiltration [50]. Also, ELANE, a neutrophil elastase, a major anticancer protein enabling neutrophils to specifically kill tumor cells, was downregulated [42].

Tumor microenvironment (TME) is becoming an increasingly interesting niche for discovering new predictive and prognostic biomarkers, as well as new mechanisms for tumor therapy development [51]. The profile of the cells infiltrating the TME, which obviously could be influenced by the expression profile of tumor tissue cells, was also in the scope of our study. We identified four subgroups of patients based on the profile of immune cells infiltrating tumor tissue. Considering the predominantly infiltrated type of cells, subtypes 1 and 2 were categorized as less immunogenic due to the lower levels of the NK cells (NK^bright^ cells subtype) and follicular helper T-cell infiltration (Tfh). The degree of infiltration of tumor tissue with NK cells is associated with survival in patients with solid tumors, but the data are not consistent—presence in many solid tumors is considered favorable, while in some others not, very likely due to their functional impairment by soluble modulators secreted in TME [52]. NK^bright^ cells represent a subset of NK cells, stratified based on CD56 expression level, preferentially recruited to the tumor site but exhibit poor cytolytic function [53]. However, it has been found that better postoperative OS in SqCLC patients is associated with an increased number of infiltrated NK cells [54]. Furthermore, Tfh cells function as an essential helper in B cell activation for effective antibody-mediated immune response. It is well described that in most solid, nonlymphatic tumors, increased infiltration of the Tfh cells correlates with a better immune response against cancer and improved clinical outcomes [55]. On the other side, tumors classified into subtypes 3 and 4 exhibit higher infiltration of CD8+ T lymphocytes in general, as well as effector and central memory T cells, and were considered more immunogenic. The higher estimated degree of infiltration of these cells suggests potentially stronger antitumor activity. Memory T cells persist longer and are capable of vigorous proliferation [56]. Effector T cells are important mediators of tumor protection; however, they are not sufficient for tumor rejection because of apoptosis-induced clonal contraction. Several clinical and preclinical studies indicate that memory cells could be more important in cancer curative immunity [57]. A recently published study showed that an insufficient number of central memory T cells in NSCLC cannot induce an adequate antitumor immune response and kill tumor cells, which may partially explain the development of refractory tumors [58].

As part of our study, the association between gene expression levels and the overall survival of SqCLC patients was also examined in nonmetastatic patients. Our analysis revealed that 36 overexpressed genes were correlated with patients’ OS. HOXC4 overexpression exhibits the highest effect on patients’ OS. Homeobox (HOX) genes encode transcription factors, and their aberrant expression affects the processes involved in tumorigenesis, such as proliferation, apoptosis, migration, and invasiveness [59]. In pediatric glioma tissues, overexpression of HOX genes in general is associated with favorable prognosis, higher immune infiltration, and better response to immunotherapy [60]. Abnormal overexpression of HOXC4 in different types of cancer (pancancer study), including SqCLC, suggests that HOXC4 may function as an oncogene and could be used as a diagnostic and prognostic biomarker [61]. Finally, we have identified the GSTZ1 gene to be associated with better OS in SqCLC patients. GSTZ1 is a member of the glutathione transferases (GST) superfamily. GST exhibits multiple biological activities, including cell protection against oxidative stress and involvement in the resistance to anticancer drugs. In cancer cells, they are often upregulated and may contribute to cell detoxification [62]. GSTZ1 is an enzyme that participates in phenylalanine/tyrosine catabolism and is frequently deregulated in cancers; however, its role in tumorigenesis is largely unknown. Li et al. found that GSTZ1 was downregulated in hepatocellular carcinoma (HCC) and associated with poor prognosis, indicating that GSTZ1 serves as a tumor suppressor in HCC [63]. We were not able to find any published data referring to the association of GSTZ1 and OS in lung cancer patients of any subtype. Therefore, our results of the GSTZ1 association with SqCLC nonmetastatic patient’s OS are novel and interesting.

Finally, due to the clear limitations of our study, our results should be cautiously interpreted. First, this study included a small number of samples (experimental cohort) and did not allow us to perform a satisfactory statistical analysis. We tried to compensate for this limitation by introducing a validation cohort. However, we did not further test any of our results in vitro; hence, the results may lead to false interpretation. Therefore, our study could be interpreted/considered as a proof of principle study. Furthermore, we did not use paired “healthy tissues” for disease-specific differential gene expression analysis but autopsy-obtained healthy lung tissues. We are aware that this approach is less common, but it has been found that cancer can systematically influence gene expression of the neighboring tissue. Several systematic molecular differences have been identified as related to immune cell activation, p38 signaling, autophagy, and reorganization of extracellular matrix, and even more interesting, molecular targets of many cancer drugs were shown to be either over- or under-represented [64]. Therefore, we found it more appropriate to include autopsy-obtained controls, although we were aware that a smaller number of controls would contribute to the variability of the results.

## 5. Conclusions

We identified a set of up- and downregulated genes specific for the SqCLC tumors and determined their functional annotations—upregulated genes mainly participate in the processes involved in the control of the cell cycle. In contrast, downregulated genes are mostly involved in the regulation of the immune response. Our study brings additional information about the transcriptomic landscape of SqCLC, uncovering potential therapeutic biomarkers and pathways. We confirmed that already recognized potential therapeutic biomarkers, MAGEA9B and MAGED4, could be interesting new targets for SqCLC. The validation across cohorts strengthens the findings, providing a foundation for future investigations. The identified genes associated with OS, HOXC4 (poor OS), and GSTZ1 (better OS), hold promise for prognostic stratification in SqCLC patients, fostering a more personalized approach to treatment. Further functional studies should elucidate which of the indicated biomarkers is present in circulation and suitable for detection by liquid biopsy, especially in the context of disease follow-up and early recurrence detection.

## Figures and Tables

**Figure 1 cancers-16-00720-f001:**
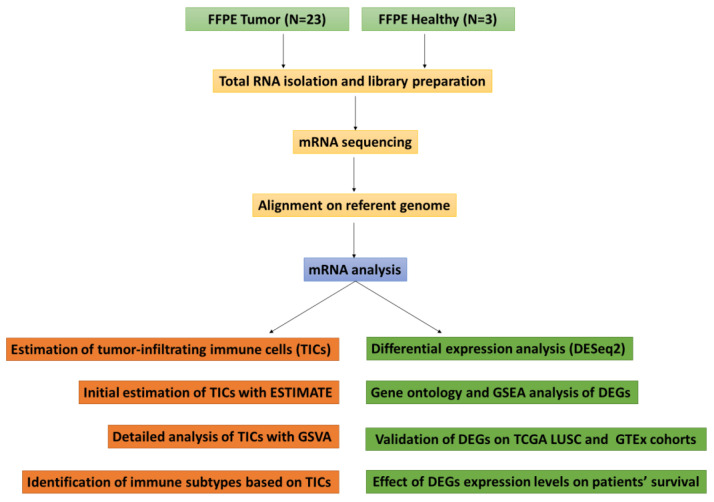
Graphical presentation of the study design.

**Figure 2 cancers-16-00720-f002:**
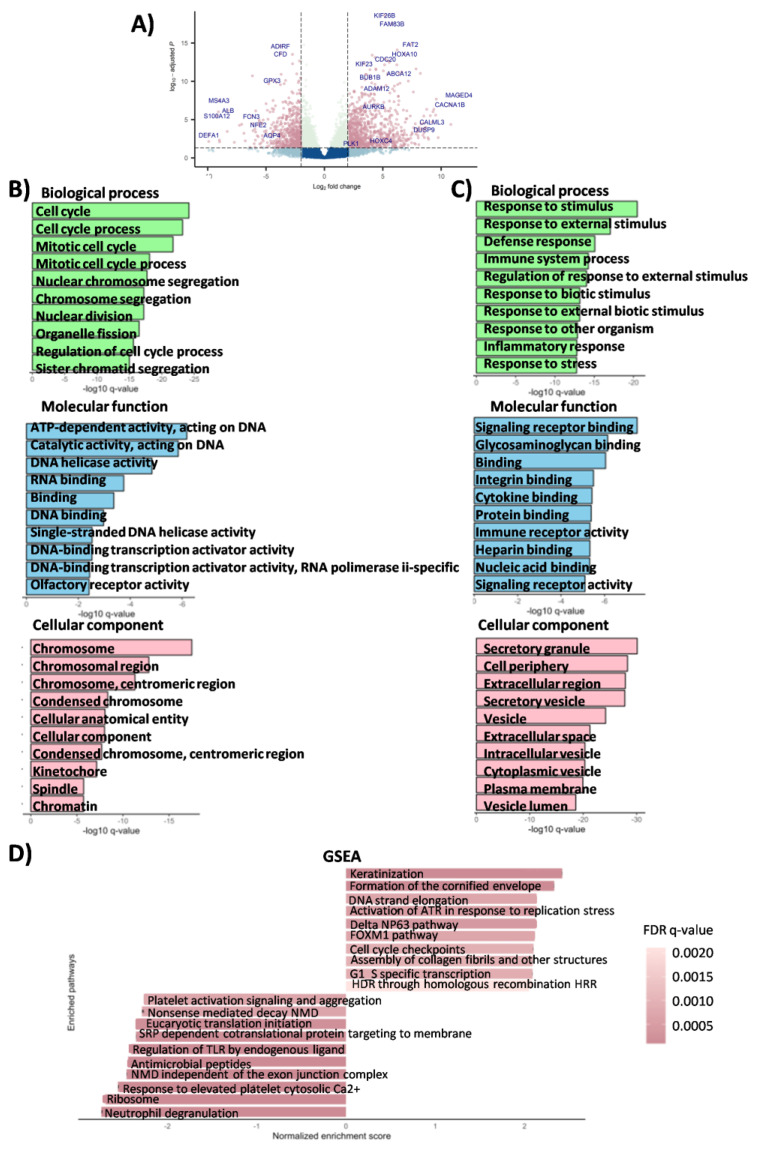
Results of the transcriptomic analysis obtained in the experimental cohort. (**A**) Volcano plot displaying differentially expressed genes (DEGs) in the experimental cohort of the SqCLC samples. Genes significantly differentially expressed in SqCLC compared to healthy controls are highlighted in pink. (**B**,**C**) Gene Ontology analysis (GO) for up- and downregulated genes in categories of Biological Process, Molecular Function, and Cellular Component (experimental cohort). (**D**) Enriched pathways in SqCLC tissue using Gene Set Enrichment Analysis (GSEA). The top 20 positively and the top 20 negatively enriched pathways are presented. The dot color indicates FDR q-values, with darker colors indicating higher significance.

**Figure 3 cancers-16-00720-f003:**
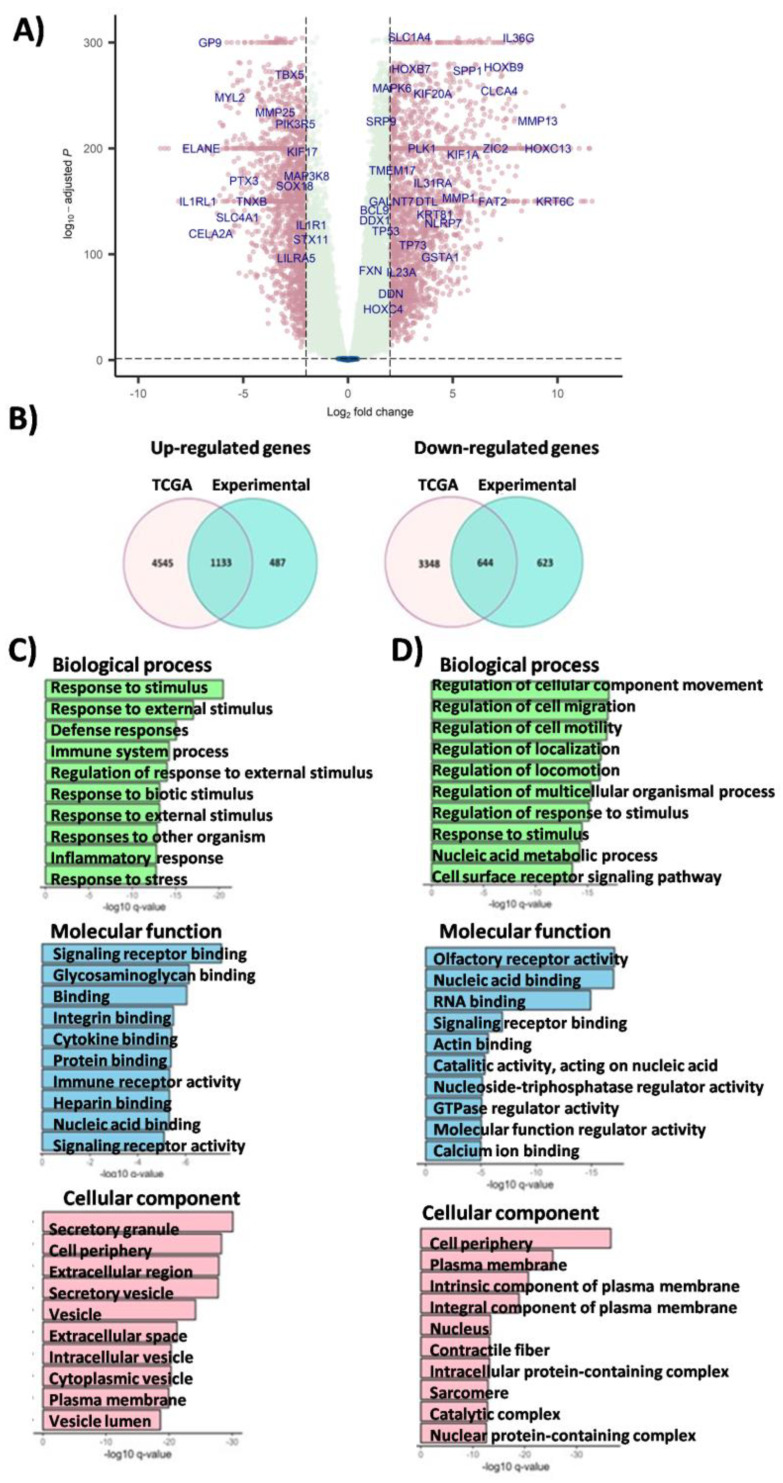
Results of the validation analysis obtained in the cohort of TCGA and GTEx samples. (**A**) Volcano plot displaying differentially expressed genes (DEGs) in the validation cohort of the SqCLC samples. Genes significantly differentially expressed in SqCLC compared to healthy controls are highlighted in pink. (**B**) Venn diagrams of commonly up- and downregulated genes in tested cohorts. (**C**,**D**) Gene Ontology analysis (GO) for up- (**C**) and downregulated (**D**) genes in categories of Biological Process, Molecular Function, and Cellular Component (validation cohort).

**Figure 4 cancers-16-00720-f004:**
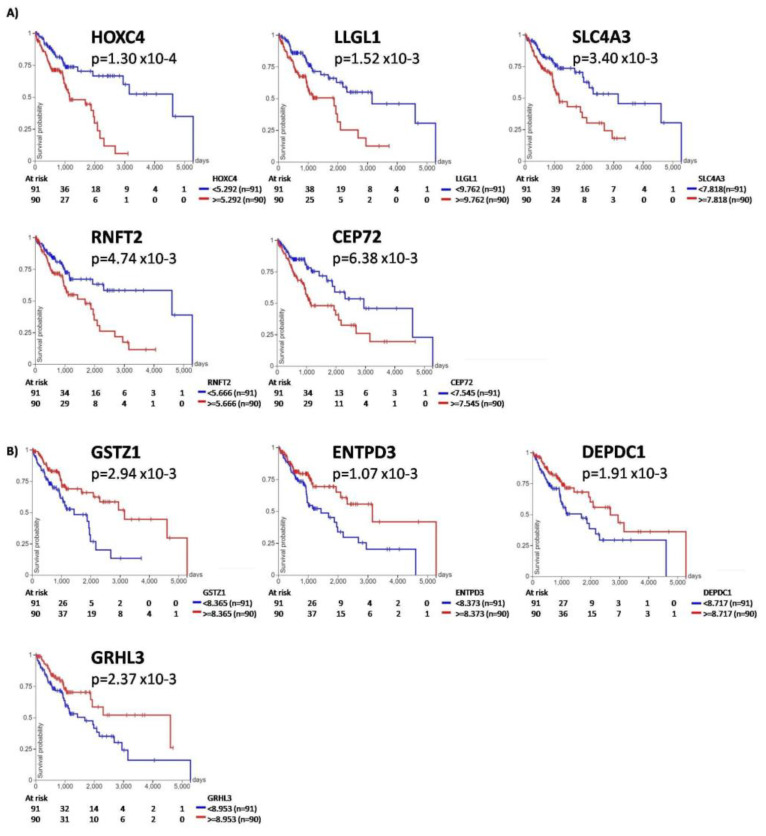
Kaplan–Meier survival curve analysis in early-stage TCGA-LUSC cohort. (**A**) Kaplan–Meier survival curve analyses for genes associated with the worst impact on patients’ overall survival (*p* < 0.01). (**B**) Kaplan–Meier curve analysis for the genes associated with a better impact on patients’ overall survival (*p* < 0.01). Patients were grouped by the median gene expression level. Red lines represent expression higher than the median, and blue lines represent gene expression lower than the median. *P*-value was calculated using a log-rank test; KM plots were generated using the Xena platform.

**Figure 5 cancers-16-00720-f005:**
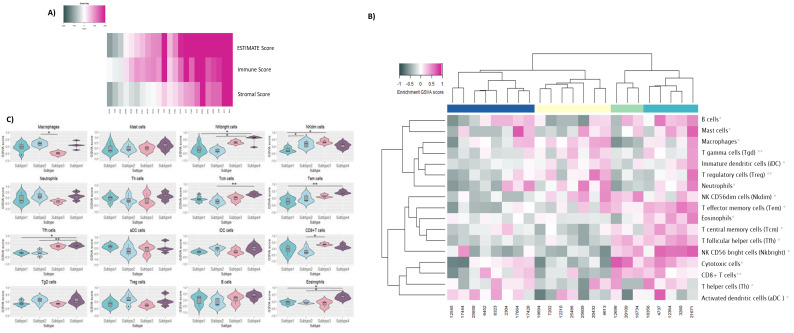
Estimation of immune cell infiltration in experimental cohort. (**A**) Estimation of immune cell infiltration and stroma presence in SqCLC samples using ESTIMATE algorithm. Grey color represents lower calculated values; pink color represents higher calculated values. (**B**) Identification of immune subtypes (S1–S4) using GSVA on 16 immune-related gene sets and hierarchical clustering. (**C**) Violin-plot presentation of different levels of immune cells infiltrating tumor in identified immune subtypes. *p*-values < 0.05 were considered significant. * *p* = 0.05–0.01; ** *p* = 0.01–0.001.

**Table 1 cancers-16-00720-t001:** Demographic and clinical characteristics of the tested SqCLC population.

Cohort	Experimental	Validation
Cases	*n* = 23	*n* = 225
Age, year (mean)	64	67
Sex (N, %)		
Male	18 (78)	163 (72.5)
Female	5 (22)	62 (27.6)
Smoking status (N, %)		
Active		75 (33.3)
Ex-smoker	9 (39)	133 (59.1)
Nonsmoker	13 (57)	8 (3.6)
Undetermined	1 (4.)	9 (4)
T stage (N, %)		
1	4 (17)	57 (25.3)
2	6 (26)	125 (55.6)
3	10 (43)	34 (15.1)
4	3 (13)	9 (4)
N stage (N, %)		
0	13 (57)	157 (69.8)
1	5 (22)	56 (24.9)
2	5 (22)	11 (4.9)
3	0	0
Undetermined	0	1 (0.4)
M stage (N, %)		
0	21 (91)	181 (80.4)
1	2 (9)	2 (0.9)
Undetermined	0	42 (18.7)

## Data Availability

Data is maintained within this article, data is not publicly available due to privacy.

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
