# Peer review of "Transcriptomic Profiling for Prognostic Biomarkers in Early-Stage Squamous Cell Lung Cancer (SqCLC)"

_cancers, 2024, doi:10.3390/cancers16040720_

Round 1

Reviewer 1 Report

Comments and Suggestions for Authors

The small sample size of your experimental group limits the accuracy to which you should express your data.  For sample size less than 100, integer accuracy of presentation only is justified.  This applies to all data in your manuscript.  Since the validation cohort is so much larger than your experimental cohort, I do not feel that expression of variance or % values beyond integer accuracy are justifiable in comparison to your experimental cohort.  I suggest you present your data accuracy only to integer values for all of your data, given the very small size of your experimental cohort.  Interpretation of your statistical analyses is limited by the extreme differences in sample size between the experimental and validation cohorts.

I think that the last paragraph of your Discussion section reflects reasonably my concerns provided in the paragraph above.

Results - Don't repeat the data in the text already presented in Figures and Tables.  Just refer to the data presentation there, and only provide a summary in the text of the findings already in the Tables and Figures.

Figures 2 -5   The font size in the body of these illustrations is too small and difficult to read.  Even with magnification on the computer screen, the present fonts were insufficiently distinct.

Given the issues raised above, your Discussion section is too long and should be substantially shortened.  It is my bias that this study is best submitted as a potential proof of principle that would justify conduct of a larger prospective study.

Author Response

Transcriptomic Profiling for Prognostic Biomarkers in Early-stage Squamous Cell Lung Cancer (SqCLC)

ANSWER TO THE REVIEWER #1 COMMENTS:

  1. The small sample size of your experimental group limits the accuracy to which you should express your data.  For a sample size less than 100, integer accuracy of presentation only is justified.  This applies to all data in your manuscript.  Since the validation cohort is much larger than your experimental cohort, I do not feel that expression of variance or % values beyond integer accuracy are justifiable compared to your experimental cohort.  I suggest you present your data accuracy only to integer values for all of your data, given the very small size of your experimental cohort.  Interpretation of your statistical analyses is limited by the extreme differences in sample size between the experimental and validation cohorts.

We understand the Reviewers’ concerns regarding the small size of our experimental group. Generally, in statistics, we could consider that a bigger sample size means higher statistical power. However, in NGS data analysis, especially RNA-Seq, there are multiple factors influencing statistical power, some of which are count distribution assumption, read length, sequencing depth, count dispersion, and pre-defined cut-offs while calculating power. Only a few papers are trying to discuss the issue of the minimal sample size needed for an NGS project, however, there is still no consensus on this issue. So far, there are only a few available tools to calculate the needed sample size to reach certain statistical power, and they could give very different numbers. In other words, there is still no scientific consensus on a minimal number of samples that should be used for specific RNA-Seq analysis pipelines. In our experiments, we used FFPE tissues from both patients with lung cancer and FFPE tissues from healthy individuals. Since healthy lung tissue is not easily available, we were able to use only what we could gather. We do agree that adding more samples in control group could increase statistical power and we are aware that smaller sample size in control group could affect Type 1. error, and to account for that we have decided to use validation cohort with sufficient number of samples in each group. Regarding presentation of data as integer values, we haven’t seen papers presenting data this way (in transcriptome analysis), even in projects using as little as few samples, and therefore we would prefer to maintain the way the data are presented in the manuscript.

  1. Results - Don't repeat the data in the text already presented in Figures and Tables.  Just refer to the data presentation there, and only provide a summary in the text of the findings already in the Tables and Figures.

We appreciate Reviewers’ comment on data presentation in Results paragraph, and have made changes to the text of the Results section accordingly.

  1. Figures 2 -5   The font size in the body of these illustrations is too small and difficult to read.  Even with magnification on the computer screen, the present fonts were insufficiently distinct.

We appreciate Reviewers’ concerns on the quality of the pictures and font sizes. We have enlarged font size of Figures 2-5 to make them more easily readable without magnification.

  1. Given the issues raised above, your Discussion section is too long and should be substantially shortened.  It is my bias that this study is best submitted as a potential proof of principle that would justify conduct of a larger prospective study.

We appreciate Reviewers’ comment. To make Discussion part of the manuscript more readable, we have substantially shortened it and added in the text that study is a potential proof of principle.

Reviewer 2 Report

Comments and Suggestions for Authors

The manuscript is quite long. Is it possible to summarize it in the methods paragraph, and report of other Aas's results?  May you in perspective indicate some treatment indications?

Author Response

Transcriptomic Profiling for Prognostic Biomarkers in Early-stage Squamous Cell Lung Cancer (SqCLC)

ANSWER TO THE REVIEWER #2 COMMENTS:

MINOR

Comment: The manuscript is quite long. Is it possible to summarize it in the methods paragraph, and report of other Aas's results?  May you in perspective indicate some treatment indications?

We appreciate Reviewer’s comment. We have shortened the Material and Methods paragraph as much as possible to retain valid information for readers.

Round 2

Reviewer 1 Report

Comments and Suggestions for Authors

1.  Formatting issues in the Introduction.  Need spaces between some words

2. Table 1 - Since sample size for the Control group is less than 100, data for this group must not be expressed beyond the accuracy of data collected.  This applies especially to volunteer ages that are not clinically relevant beyond integer presentation, even for sample sizes greater than 100.  Variance is not to be expressed to a greater accuracy than the mean for the sample group.  This concern remains for data presentation for all presentation in the control group.  For some data in the validation group, excepting for age, data expression to one decimal place is justifiable. 

Comment - Just because  orthers do not report data correctly does not justify letting this type of data reporting persist in the current manuscript.

Results

Section 3.1  The authors have not removed redundant presentation of data already provided in Table 1, even though they have claimed to have remedied this issue in their response to this reviewer's concerns. Please just interpret the data already provided in Table 1.

I do not find a copy of Figure 2 in the revised manuscript - only the legend to it provided in lines 341 - 7.

Figure 3, Panel A is still to small to be read.  Please enlarge the presentation to make it reasonable to read and interpret by the average reader of this paper.

Other statistical presentation in the Results section is now appropriate.

I am not convinced that the Discussion section has been reduced in size as recommend by both reviewers, although the issue of proof of principle is now adequately addressed.

Author Response

Transcriptomic Profiling for Prognostic Biomarkers in Early-stage Squamous Cell Lung Cancer (SqCLC)

Dear,

Thank you for conveying your decision on the above manuscript to us.  Again, we would like to express our sincere appreciation to the Reviewer for his hard work and constructive comments on a revised version of our manuscript. We fully understand all of his comments, we accept them all and we have addressed them in the revised version. There, all changes suggested by the Reviewer are incorporated using the track changes. In our point-by-point reply to each of the Reviewer's comments, we hope to demonstrate that the Reviewers’ concerns have been met in the revised version. We hope that with these changes our manuscript will be acceptable for publication in Cancers.

Sincerely Yours.

Jelena Knežević, PhD

1.Formatting issues in the Introduction.  Need spaces between some words. 

We are thankful for the comment and we believe that in the revised version of the manuscript, all spaces are correctly placed. We apologize if there are still some mistakes that could be corrected later, in case that manuscript will be accepted.  

2. Table 1 - Since sample size for the Control group is less than 100, data for this group must not be expressed beyond the accuracy of data collected.  This applies especially to volunteer ages that are not clinically relevant beyond integer presentation, even for sample sizes greater than 100.  Variance is not to be expressed to a greater accuracy than the mean for the sample group.  This concern remains for data presentation for all presentation in the control group.  For some data in the validation group, excepting for age, data expression to one decimal place is justifiable.  Comment - Just because  orthers do not report data correctly does not justify letting this type of data reporting persist in the current manuscript. 

We are thankful for this comment and the revised version of the manuscript is corrected accordingly. We agree that mistakes made by others should not be repeated and appreciate this comment. 

3. Results - Section 3.1  The authors have not removed redundant presentation of data already provided in Table 1, even though they have claimed to have remedied this issue in their response to this reviewer's concerns. Please just interpret the data already provided in Table 1. 

In the revised version of the manuscript, we try to remove as much as possible redundant data/results. We belive that new version is more appropriate.

I do not find a copy of Figure 2 in the revised manuscript - only the legend to it provided in lines 341 - 7.

We apologize for this mistake, but during the revision process somehow we lost Figure 2 - it should be now incorporated.

Figure 3, Panel A is still to small to be read.  Please enlarge the presentation to make it reasonable to read and interpret by the average reader of this paper. Other statistical presentation in the Results section is now appropriate.

We accepted the Reviewer's comment and enlarged the font size - we hope that the new version of Figure 3 will meet the reviewer criteria.

4. I am not convinced that the Discussion section has been reduced in size as recommend by both reviewers, although the issue of proof of principle is now adequately addressed. 

We agreed with both of the Reviewers that the initial version of the Discussion part could be more condensed.  Therefore, we accepted their suggestion and deleted as much as possible of the discussed parts of the text, however, we still did not want to miss some important parts. Please, refer to the pdf file, that was resubmitted in the first round of the process, where you can see all parts that were deleted. 

Round 3

Reviewer 1 Report

Comments and Suggestions for Authors

Thank you for addressing the issues identified in my second review of this manuscript